# Kinase-Independent Functions of MASTL in Cancer: A New Perspective on MASTL Targeting

**DOI:** 10.3390/cells9071624

**Published:** 2020-07-06

**Authors:** James Ronald William Conway, Elisa Närvä, Maria Emilia Taskinen, Johanna Ivaska

**Affiliations:** 1Turku Bioscience Centre, University of Turku and Åbo Akademi University, 20520 Turku, Finland; james.conway@utu.fi (J.R.W.C.); elisa.narva@utu.fi (E.N.); maria.taskinen@utu.fi (M.E.T.); 2Department of Biochemistry, University of Turku, 20520 Turku, Finland

**Keywords:** MASTL, actin, contractility, cell cycle, cancer, therapeutic targeting, kinase inhibitor

## Abstract

Microtubule-associated serine/threonine kinase-like (MASTL; Greatwall) is a well-characterized kinase, whose catalytic role has been extensively studied in relation to cell-cycle acceleration. Importantly, MASTL has been implicated to play a substantial role in cancer progression and subsequent studies have shown that MASTL is a significant regulator of the cellular actomyosin cytoskeleton. Several kinases have non-catalytic properties, which are essential or even sufficient for their functions. Likewise, MASTL functions have been attributed both to kinase-dependent phosphorylation of downstream substrates, but also to kinase-independent regulation of the actomyosin contractile machinery. In this review, we aimed to highlight the catalytic and non-catalytic roles of MASTL in proliferation, migration, and invasion. Further, we discussed the implications of this dual role for therapeutic design.

## 1. Introduction

The serine/threonine kinase MASTL (microtubule-associated serine/threonine kinase-like; Greatwall) was first described in *Drosophila melanogaster* for the role it played in supporting cell-cycle progression [1]. Importantly, the *MASTL* gene is highly conserved from arthropods to vertebrates, speaking to the essential part that it plays in both development and tissue homeostasis [2,3] (Figure 1A,B). In mammals, homozygous loss of MASTL is embryonically lethal at embryonic day 10.5, suggestive of pre-implantation lethality and highlighting the essential role of MASTL during development [4]. 

The best-known roles for MASTL are through kinase-dependent regulation of mitosis and meiosis, but several kinase-independent functions for MASTL during migration, adhesion, and invasion have recently been described (Figure 2). Many kinases have important functions independent of their kinase domains, including scaffolding, subcellular targeting, and direct or indirect DNA binding as a cofactor for transcription, along with allosteric and competitive roles for other enzymes [7]. For example, the primary phosphorylation substrate for focal adhesion kinase seems to be itself, where many of its functions are attributed to its ability to scaffold and recruit adhesion and signaling components. Similarly, many members of the cyclin family have been recorded to have kinase-independent functions in transcriptional regulation [8]. This suggests that many kinases may in fact sit within far more complex regulatory networks, working beyond post-translational regulation of substrates. Here we will discuss recent studies that support a role for MASTL independent of its kinase activity, outlining the dual functionalities and attempting to delineate the kinase-dependent and -independent roles during cancer progression.

## 2. MASTL in Cancer

Cancer is responsible for the highest disease burden globally [9], and MASTL is implicated as a poor prognostic factor in several of the most lethal cancer subtypes, including breast [10,11], gastric [12], colon [13], liver, non-small-cell lung cancer (NSCLC), and ovarian (Figure 3). The roles of MASTL in cancer are many and various, where multiple studies have reported that silencing decreases cell proliferation, migration, and invasion, while overexpression can enhance these properties (Figure 3A). Furthermore, MASTL has recently been shown to inhibit cell spreading and attachment to the extracellular matrix (ECM) [14], modify cell–cell contacts [14,15], and reduce cell–cell contact inhibition [15,16]. Importantly, genetic ablation of MASTL has a significant therapeutic effect in vivo [11,13,15,16,17], and this provides a solid basis for further therapeutic investigation.

### 2.1. Possible Kinase-Dependent Functions of MASTL in Cancer

The best-characterized substrates of MASTL kinase are cAMP-regulated phosphoprotein 19 (Arpp19) and endosulfine alpha (ENSA, identified due to its high homology with Arpp19), both of which contain the conserved MASTL phosphorylation motif FDSGDY [21,22]. This kinase function of MASTL has primarily been associated with cell-cycle progression (reviewed in [23]), where it facilitates effective mitotic exit through downregulation of the B55 subunit of protein phosphatase 2A (PP2A) [24]. This prevents excessive dephosphorylation by PP2A and mitotic collapse upon nuclear envelope breakdown [4]. Phosphorylation of ENSA by MASTL is important for this process, as this allows ENSA to inactivate the B55 subunit [25]. Furthermore, loss of MASTL activity can lead to mitotic slippage in response to DNA damage due to a retained activity of PP2A-B55 and dephosphorylation of polo-like kinase 1 (Plk1) [26]. Interestingly, knockout *Xenopus* models for Arpp19 or ENSA highlight their essential nature in separate phases of the cell cycle, where ENSA loss perturbs synthesis phase (S phase) progression, while Arpp19 was found to be more important for mitotic phase (M phase) [22]. Together, the combined role of MASTL during proliferation is highlighted by a clear defect when MASTL is knocked out, but this could be restored with wild-type and not kinase-dead G43S (murine mutant corresponding to G44S in human, Figure 1a), implying that these cell-cycle effects are kinase-dependent [17]. Collectively, the essential functions of MASTL in maintenance of a stable genome and correct progression through the cell cycle highlight the importance of the kinase activity of this multifunctional protein. 

#### 2.1.1. Glycogen Synthase Kinase 3 (GSK3), AKT, and Wnt Activity

Recent work has linked glycogen synthase kinase 3 (GSK3) activity with the invasive properties of MASTL [16]. In this work, the authors found that MASTL overexpression was paralleled by an increase in phosphorylation of the protein kinase B (PKB, also known as AKT) [16], which is known to regulate multiple proteins involved in actin cytoskeletal polymerization and stabilization [27]. MASTL was found to increase GSK3 activity, which is known to phosphorylate PH domain leucine-rich repeat protein phosphatase (PHLPP), resulting in proteasomal degradation [16]. As PHLPP is able to inactivate AKT through dephosphorylation, loss of PHLPP led to an increase in AKT phosphorylation and downstream activation. However, none of these proteins are direct targets of MASTL kinase, leaving the direct mechanism open for investigation. Another study performed global phosphoproteomic analyses of immortalized breast MCF10A cells engineered to overexpress MASTL [15]. Although AKT phosphorylation could not be detected by phosphoproteomics, a slight increase of AKT S473 phosphorylation was observed using an antibody-based phosphoprotein screen, along with a significant increase in mechanistic target of rapamycin complex 1 (mTORC1) activity. Conversely to the previous study, significant inactivation of GSK3 was measured upon MASTL overexpression. Further to this complexity, MASTL silencing in colon cancer cells (HCT116 and SW620) has been shown to activate GSK3, leading to a marked decrease in the β-catenin nuclear accumulation and Wnt-/β-catenin activity [13]. β-catenin plays a critical role in divergent cellular processes, including promotion of migration and invasion [28,29]. In brief, MASTL levels seem to affect AKT and GSK3 phosphorylation, but there may be context-dependent variation and the detailed mechanism and possible kinase targets of MASTL in these pathways remains unclear.

#### 2.1.2. Phosphorylation of Cytoskeletal- and Adhesion-Related Proteins

Multiple studies have indicated a clear phenotype for MASTL in regulation of the actin cytoskeleton. Overexpression in immortalized MCF10A cells has been reported to lead to disorganization of the actin network and induction of a partial epithelial-to-mesenchymal transition (EMT) [15]. Subsequent phosphoproteomic screening identified increased phosphorylation of sciellin (SCEL) and epiplakin 1 (EPPK1), which are linked to actin cytoskeletal dynamics. Further phosphoproteomic analysis, in cells overexpressing a clinically relevant MASTL-mutant (E166D murine, E167D human Figure 1a, see below) indicated hyperphosphorylation of focal adhesion- and cytoskeleton-associated proteins, namely exostosin like glycosyltransferase 3 (Diap1), dedicator of cytokinesis 1 (Dock1), Filamin A (Flna), p21 RAC1-activated kinase (Pak1-3), phosphoinositide-3-kinase regulatory subunit 1 (Pik3r1), protein kinase C alpha (Prkca), raf1 proto-oncogene (Raf1), talin 1 (Tln1), zyxin (Zyx), mitogen-activated protein kinase kinase 2 (Map2k2), myosin heavy chain 9 (Myh9), wasp family member 2 (Wasf2), vasodilator stimulated phosphoprotein (Vasp), and vinculin (Vcl) [30]. However, none of these studies was able to show that the identified phosphorylation sites were direct targets of MASTL. Interestingly, a recent proteomics approach assessed the role of the non-conserved middle region (NCMR) of MASTL and found that this was crucial for target specificity and kinase activity (Figure 1a, [5]). The specific and direct phosphoproteomic assay in HEK293 cells revealed phosphorylation of actin gamma 1 (ACTG1), Talin (TLN1), and phosphatidylinositol-4-phosphate 5-kinase (PIP5K1C). Of note, this study revealed a cancer-associated truncated version of MASTL, referred to as MASTL_450_ (Figure 1a) that displays catalytic activity, not on Arpp19 or ENSA, but on novel substrates, such as high mobility group protein A1 (HMGA1), which has been linked to the metastatic progression of cancer cells. Importantly, no proteomic studies have yet assessed the changes associated with MASTL overexpression on a kinase-dead background, such as G44S, where a greater clarity could be achieved as to the dependence of these phosphorylation changes on direct MASTL targeting. Furthermore, many immunohistochemical or RNAseq datasets looking at the influence of MASTL on cancer progression have not considered the implications of MASTL isoforms, and this may be an area for future investigation or data mining. 

An interesting clinical mutation in MASTL is E167D (Figure 1A), which has been identified in patients with autosomal dominant inherited thrombocytopenia. This mutation occurs in the kinase domain of MASTL, which has a high homology with AGC kinases [31], but differs in the NCMR [5,32]. This mutation was initially thought to inhibit kinase function, but subsequent studies have suggested that it, in fact, increases the association with PP2A-B55 [30]. To model this mutation in vivo, a zebrafish system was applied that utilized a transient morpholino knockdown of Mastl [33]. This identified a dramatic role for MASTL in the maintenance of hematopoietic progenitors, supporting the clinical findings. Importantly, the thrombocytopenia-associated Mastl knockin E166D mutation has been shown to deregulate actin cytoskeletal dynamics during platelet activation, while both Mastl-deficient (Δ/Δ) and E166D-mutant (ED/ED) platelets showed a defect in the generation of actin fibers and lamellipodia after stimulation with fibrinogen [30]. This supports the phosphoproteomics studies in that the identified actomyosin substrates are likely having a functional effect in these models, which is disrupted by the E167D mutation. 

### 2.2. Possible Kinase-Independent Effects of MASTL on the Actomyosin Cytoskeleton 

MASTL has also been linked with increased motility, invasiveness, and metastatic progression in breast [14,15] and gastric cancers [12]. In our recent work [14], we find that MASTL inhibits cell spreading and attachment. We also found that MASTL depletion influenced the transcriptome and proteome of breast cancer cells, affecting genes implicated in cell movement and actomyosin contraction. Mechanistically, MASTL associated with myocardin related transcription factor A MRTF-A and increased its nuclear retention and transcriptional activity, leading to increased expression of MRTF-A target genes tropomyosin 4.2 (*TPM4*), vinculin (*VCL*), and non-muscle myosin IIB (NM-2B, *MYH10*). In addition, MASTL supported expression of Rho guanine nucleotide exchange factor 2 (GEF-H1, *ARHGEF2*), which is not an MRTF-A target gene, but, in fact, induces MRTF-A activity [34]. Of note, all of these proteins have important roles in actomyosin contraction and migration, and rescue with a kinase-inactive G44S human mutant MASTL was sufficient to restore these phenotypes [14]. This implies that the effect of MASTL on cell spreading and attachment to the extracellular matrix is kinase-independent. Given the established link between MASTL and metastasis in both breast and gastric cancers, it seems likely that this role may also be kinase-independent. This is an essential aspect for future clinical targeting, where kinase inhibitors may be met with less-than-adequate results without proper consideration.

MASTL has recently been shown to regulate actin cytoskeletal architecture in interphase normal mammary epithelial and breast cancer cells [14] and in post-mitotic platelets [30]. Dynamic regulation of the actin cytoskeleton is essential for different steps of cell division, such as cell rounding, separation of centrosomes, and formation of a contractile ring during cytokinesis, reviewed in [35,36,37]. This suggests a possible role for MASTL as a regulator of actin re-organization and contraction during different steps of mitosis and cytokinesis, but this has yet to be assessed. Interestingly, many of the components MASTL regulates during interphase, including non-muscle myosin II, GEF-H1, and MRTF-A [14], are also involved in regulation of the cell cycle [38,39,40,41,42,43]. This raises the question that, in addition to the well-established MASTL-ARPP19-ENSA-PP2A-B55 pathway (reviewed in [10,23], could MASTL regulate mitosis also via actin remodeling? Since MASTL can regulate non-muscle myosin IIB expression and myosin activity by increasing phosphorylation of myosin light chain in interphase breast cancer cells, it would be interesting to study if MASTL could affect cell rounding via myosin activation. 

GEF-H1 has been demonstrated to be an important regulator of cell cytoskeletal organization, actomyosin contraction, and cell-cycle progression in multiple normal and malignant cell types [40] and it facilitates normal mitotic spindle assembly and mitotic progression in fibroblasts, kidney epithelial, and cervical cancer HeLa cells [38]. During early mitosis, GEF-H1 is phosphorylated and, thus, inhibited by Aurora A/B and Cdk1/Cyclin B. In contrast, prior to exit from mitosis, GEF-H1 is activated by dephosphorylation, after which it is able to activate RhoA and increase contractility at the cleavage furrow in HeLa cells [39]. However, the dependency of RhoA activation during cytokinesis is cell-type specific; C3-mediated RhoA inhibition blocks cytokinesis in HeLa, but not in Rat1 fibroblasts [44]. As described earlier, MASTL regulates actomyosin contraction through GEF-H1 during interphase, but it remains to be studied if MASTL could affect GEF-H1 during mitosis and/or cytokinesis. In addition, MASTL has been shown to regulate destruction of Cyclin B, enabling successful anaphase in diploid-immortalized human retinal pigment epithelial cells and several cancer cells, including osteosarcoma (U2OS), colon (HCT116) and cervical cancer (HeLa) [45], while demonstrating a possible stabilizing effect on Aurora A levels in neuroblastoma patient samples [46]. These findings suggest that MASTL might affect GEF-H1 activity via Cyclin B and Aurora A during mitosis. Furthermore, MRTFs have a role in timely cell-cycle progression and chromosomal stability in fibroblasts, and depletion of MRTF-A/B leads to shortened G1 phase and lengthened S and G2 phases. In this case, the fibroblasts tend to become micronucleated, aneuploid, or polyploid after MRTF silencing [43]. To clarify the kinase-independent role of MASTL in the cell cycle, through MRTF-A association and GEF-H1 activation, future studies would benefit from careful assessment of the cell cycle during mitosis, as well as utilizing kinase-active and -inactive constructs to maximize the clarity of the effect.

## 3. MASTL as a Therapeutic Target

While an increasing number of studies are demonstrating a clear benefit from MASTL inhibition, in relation to reducing chemoresistance, radioresistance, metastatic progression, and overall survival (Figure 3A), there remains a lack of effective inhibitors for this multi-functional protein. Direct inhibition of MASTL requires a compound to penetrate the plasma membrane, reducing the efficacy of monoclonal antibodies as therapeutics [47]. While a greater understanding of MASTL targets could yield downstream therapeutic opportunities, one study has assessed the direct inhibition of the MASTL kinase domain (Figure 4) [48]. Here the authors used structure–activity relationships and a crystal structure from a reduced kinase construct to improve compounds from a small-molecule screen. This approach yielded a highly active compound against the MASTL kinase functions, but maintained some off-target effects against other AGC kinases, including Rho-associated protein kinase 1 (ROCK1). Several in silico studies have investigated the perturbation of MASTL through a network biology approach, highlighting a potential therapeutic benefit from MASTL inhibition in neuroblastoma [46] and supporting investigations in breast cancer [15]. This has been taken one step further, where in silico predictions for MASTL kinase inhibitors identified several naturally occurring compounds [49]. This virtual screening approach is an emerging area for clinical development, where compounds can be run against crystal structures of the kinase domain to narrow down lists and reduce the lengthy screening processes that currently incur excessive costs for the pharmaceutical industry [50].

As we have shown in our recent work [14], the kinase-independent functions of MASTL also provide a tantalizing therapeutic opportunity (Figure 4). In a clinical setting however, knockdown of whole proteins presents several challenges. RNA interference (RNAi) for targeted degradation of messenger RNA uses small RNA molecules complementarily to mutated or endogenous mRNA for cancer drivers, and is commonplace in vitro, or even in vivo, but clinical trials have been slow to progress due to the challenges with targeting organs other than the liver and the associated toxicities. However, if we look beyond the idea of removing or inactivating MASTL, a more targeted approach could be to inhibit the adaptor interactions that are associated with the oncogenic role of MASTL, such as disruption of the dimerization interface responsible for the MRTF-A/MASTL complex and its functions (Figure 4). Regardless of the approach, it is clear that a more detailed understanding of the kinase-dependent or -independent functions of MASTL is an essential step in developing therapeutic approaches against this multifunctional oncogenic protein.

## 4. Conclusions

Although the role of MASTL has been widely studied in cancer, it remains unclear whether MASTL supports cancer progression via kinase-dependent or -independent functionalities (Figure 2). The role in supporting cancer proliferation is most likely dependent on the kinase activity and linked to cell-cycle progression. Currently, several approaches are being undertaken to develop specific and potent MASTL kinase inhibitors. These are necessary to fully uncover the therapeutic value and possible limitations of targeting MASTL kinase activity of cancer (Figure 4). However, cancer progression is highly dependent on cell contractility and interactions with the ECM, both of which seem to be regulated by MASTL, independent of its kinase activity. This appears to occur predominantly through its interaction with MRTF-A, suggesting that disruption of the interaction may provide an efficient strategy against MASTL-driven invasion (Figure 2, Figure 4). However, design of effective protein–protein interaction inhibitors against MASTL would require detailed information of the structure of the MRTF-A/MASTL complex, which is currently lacking. Furthermore, the kinase dependency remains completely unstudied in relation to in vivo migration, invasion, and metastasis. Future work to resolve the kinase dependency of MASTL functions should involve in vivo assessment of kinase-dead MASTL or the use of a potent and selective inhibitor. 

## Figures and Tables

**Figure 1 cells-09-01624-f001:**
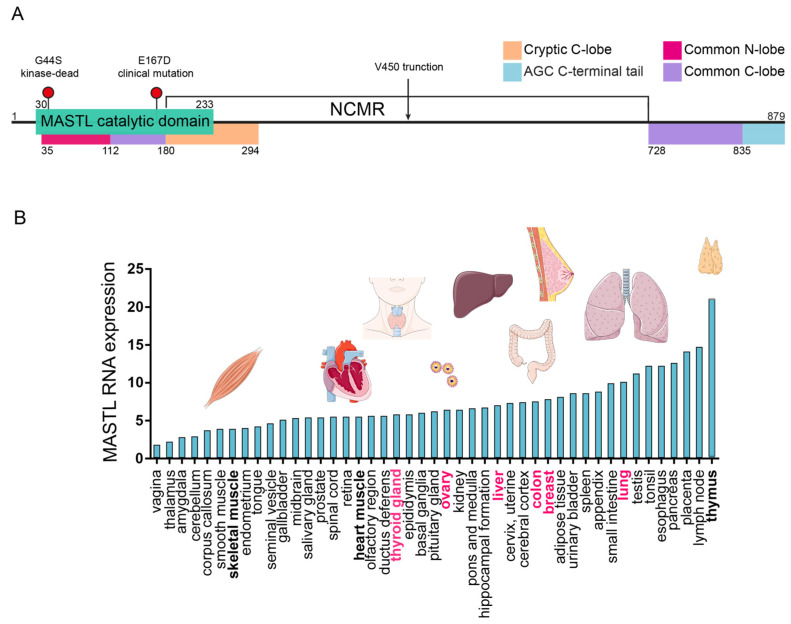
Microtubule-associated serine/threonine kinase-like MASTL structure and expression in normal tissue and cancer. (**A**) Schematic of the human MASTL gene showing the catalytic domain (green), along with the sites of kinase-inactivating (G44S) and clinically relevant (E167D) mutations. In line with Hermida, D. et al. (2020), the common C- and N-terminal lobes (purple/pink), cryptic C-lobe (orange), AGC N-terminal tail (blue), and the non-conserved middle region (NCMR) (white) are annotated. The site of the V450 cancer-relevant truncation is also indicated. For further structural details, please refer to [5]. (**B**) MASTL RNA expression in normal tissue [6] (URL: http://www.proteinatlas.org). The organs written with bold text are provided as illustrations (illustrations provided by Servier Medical Art under a Creative Commons license, URL: https://smart.servier.com). The organs (thyroid gland, ovary, liver, colon, breast, and lung) written in pink text are where high MASTL expression has been shown to correlate with poor cancer prognosis (or where MASTL has been associated with cancer).

**Figure 2 cells-09-01624-f002:**
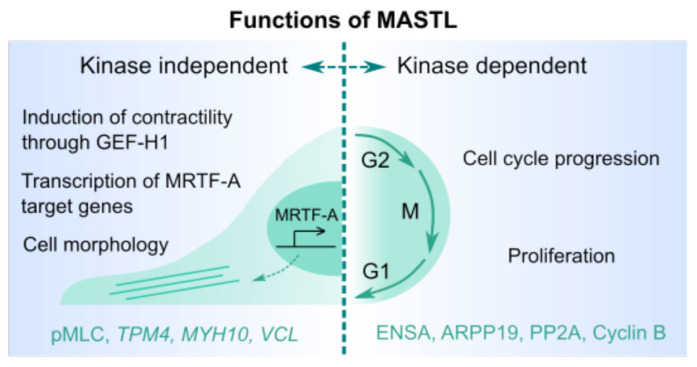
Illustration of known MASTL functions. The divisions between kinase-dependent and -independent functions are suggestive and need to be studied further.

**Figure 3 cells-09-01624-f003:**
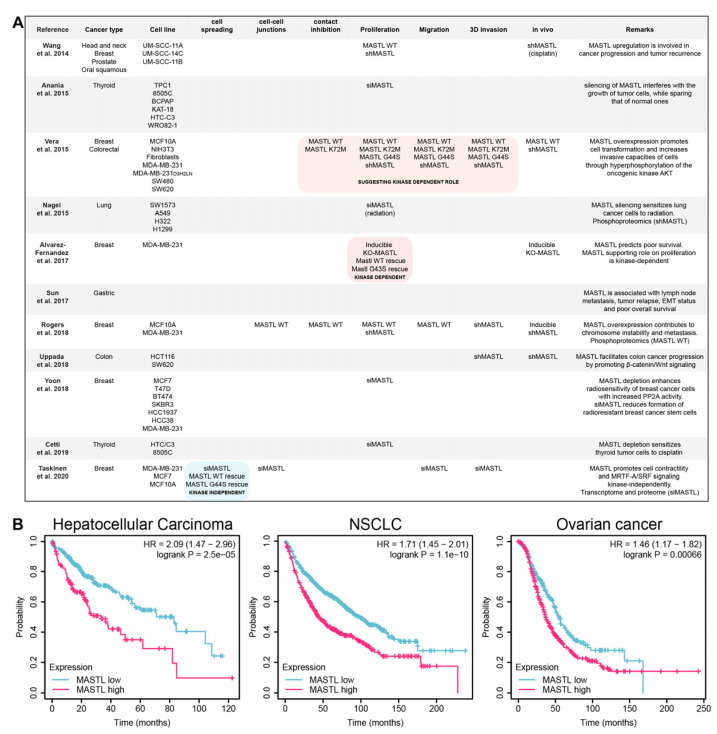
MASTL in cancer. (**A**) Literature describing MASTL function in various cancer types. The functional experiments (MASTL WT: overexpression of the wild type, MASTL K72M: overexpression of the hyperactive kinase, MASTL G44S(human)/G43S(murine): overexpression of kinase dead, siMASTL: small interfering RNA-based silencing, shMASTL: short hairpin RNA-based silencing) in relation to cell spreading, cell–cell junctions, contact inhibition, proliferation, migration, 3D invasion, and in vivo experiments are indicated. The kinase-independent (blue) and kinase-dependent (red) functions are highlighted. (**B**) MASTL expression survival analysis from RNA sequencing datasets, where plots show overall survival in hepatocellular carcinoma, non-small-cell lung cancer (NSCLC) and ovarian cancer in MASTL high or low expressing patients. Kaplan–Meier plots were generated using KMPlot (URL: http://kmplot.com; [18,19,20]).

**Figure 4 cells-09-01624-f004:**
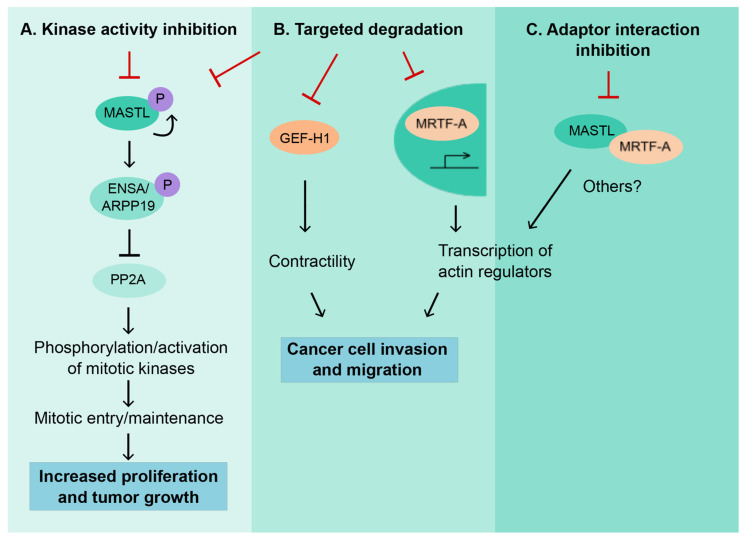
Schematic representation of the possible approaches for MASTL therapeutic targeting and the expected biological outcomes from each strategy. (**A**) Inhibition of MASTL kinase activity could lead to decreased MASTL autophosphorylation [32] and attenuation of the MASTL- Endosulfine alpha (ENSA)/ cAMP-regulated phosphoprotein 19 (ARPP19)- Protein phosphatase 2A (PP2A) pathway, cell proliferation, and tumor growth. (**B**) Targeted degradation of MASTL could lead to decreased cancer cell invasion and metastasis via reduced Rho guanine nucleotide exchange factor 2 (GEF-H1) expression and myocardin related transcription factor A (MRTF-A) activity. (**C**) Inhibition of the interaction between MASTL and MRTF-A could lead to decreased MRTF-A activity, and, thus, reduced cancer cell motility.

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
