# Peer review of "Kinase-Independent Functions of MASTL in Cancer: A New Perspective on MASTL Targeting"

_cells, 2020, doi:10.3390/cells9071624_

Round 1

Reviewer 1 Report

The review 'Kinase-independent functions of MASTL in cancer: a new perspective on MASTL targeting' discuss in detail the kinase-independent function of MASTL which important in other ways for therapeutic targets in different cancer types.

Minor Comment:

Line 46 to 55 fonts are different.

Line 213 spell check for tissue.

Major Comment:

The authors discuss the function of MASTL in a kinase-dependent or kinase-independent manner. It would be nice to show the MASTL protein structure or domains showing the kinases and other domains as well.

Author Response

The review 'Kinase-independent functions of MASTL in cancer: a new perspective on MASTL targeting' discuss in detail the kinase-independent function of MASTL which important in other ways for therapeutic targets in different cancer types.

Minor Comment:

Line 46 to 55 fonts are different.

Au: We have now corrected the discrepancy in the fonts throughout the text.

Line 213 spell check for tissue.

Au: We have now fixed the spelling for “tissue” – line 218.

Major Comment:

The authors discuss the function of MASTL in a kinase-dependent or kinase-independent manner. It would be nice to show the MASTL protein structure or domains showing the kinases and other domains as well.

Au: We thank the reviewer for the comments and have now added a new Figure 1a to illustrate the different domains for MASTL, along with the important known mutations that have functional significance. Within the figure legend, the readers are also directed towards reference 26, where detailed structural assessment of MASTL has been performed.

Reviewer 2 Report

Conway and colleagues summarized scientific evidences about MASTL catalytic and non-catalytic activities and implications for cancer therapy. 

I suggest to revise the paper for a smart read. In detail:

1) MASTL in cancer: clinical data (including a table)+Fig. 1A and 1C

2) Functions of MASTL: Kinase indipendent and dependent (Fig. 1b and table)

3) Therapeutic strategies (add a figure to summarize therapeutic hypothesis.

Fig.1A Legend should be improved

Fig. 1B Pathway and genes reported in the text could be included

Author Response

Reviewer #2:

Conway and colleagues summarized scientific evidences about MASTL catalytic and non-catalytic activities and implications for cancer therapy.

I suggest to revise the paper for a smart read. In detail:

1) MASTL in cancer: clinical data (including a table)+Fig. 1A and 1C

2) Functions of MASTL: Kinase indipendent and dependent (Fig. 1b and table)

3) Therapeutic strategies (add a figure to summarize therapeutic hypothesis.

Au: We thank the reviewer for their insightful advice. We have now restructured our headings to meet the suggestions, as well as updating the figures to fit within the suggested outline.

Fig.1A Legend should be improved

Au: Following the reviewers advice, we have now improved the legend for Fig. 1a.

Fig. 1B Pathway and genes reported in the text could be included

Au: With the additional new figures and textual revisions, this is now Figure 2. The figure has been updated to include the MASTL regulated genes and pathways mentioned in the text.

Reviewer 3 Report

Author of this manuscript reviewed recent advances in the field of the role of MASTL regulation in cancer. The authors focused on the kinase-independent functions of MASTL and the potential therapeutic application of MASTL hampering. The introduction provides background and includes relevant references. This is very interesting and prevailing topic; however it is presented in a little  disorganized way.

Some of concerns are listed below:

It is not exactly clear, which parts of MS refer to regulation of MASTL in cancer cells and which describe mechanisms in normal cells, especially in lines 163-179. Cancer cell type/origin should addressed in some sentences (the statement “in cancer cells” is not enough).

In the section “AKT, Glycogen synthase kinase 3 (GSK3) and Wnt activity” – the meaning of the presented regulation is lost – what GSK3 inactivation accounted for? And conversely, what happened in tumour cell when GSK3 was activated? What was the cause of such different regulation? Cancer cell phenotype? Authors present some interesting findings, but no explanation is given. The findings should be better addressed – how the changes in signal transduction pathways accounted for cancer cell functioning (e.g. invasive properties), phenotype? The most essential parts are missing.

Figure 2b left panel: since kinase-independent functions of MASTL in cancer was the main topic, and mechanistic studies were highlighted throughout the text, this illustration should better illustrate the presented findings. The main upstream regulators along with downstream effectors may be showed in detail. Alternatively, each mechanism the Authors describe, may be presented in a separate figure.

The manuscript needs editing and some attention to sentence structure so that the data are more clear to readers. Sections have a confusing numbering (there are two 2.1.1 sections, the first 2.1.1 section is vast and not divided into smaller fragments). Why did the authors create a separate paragraph 4.1. Figures and Tables? Figures and tables should be presented in the text, where necessary.

There are some shortcomings throughout the text: 

Line 77: “Hyperactivation of AKT S473 has been proposed..”,

Line 84: “a slight increase of AKT S473 was observed”

Line 215: „The organs (names?) written with (in?) pink…”

Please, give full name for all acronyms throughout the text, where listed for the first time, or, alternatively, put all abbreviations into one section (e.g. line  100 “cytoskeleton-associated proteins; namely Diap1, Dock1, Flna, Pak1-3, Pik3r1, Prkca, Raf1, Tln1, Zyx, Map2k2, Myh9, Wasf2, 101 Vasp, Vcl”; line 119: “NCMR ” was not defined)

Please, use Italic for “via” “in vitro”, “in vivo” throughout the text (e.g. lines 202, 227, 232, 236, 238, Table 1).

Conclusions: some of the most important facts need to be better addressed. The main observations and findings cited  in previous paragraphs should be pointed out here. Since one of the main goals of the MS was to present how targeting of MASTL may be used for anti-cancer therapy, the most important findings in the area may be presented in conclusion section.

The text need editing (two types of font were used).

The English style and grammar should be improved, especially:

Line 36:”… the primary phosphorylation substrate of focal adhesion kinase seems to be itself”

Line 49:”… The roles of MASTL in cancer are many and varied….”

Line 82: “Another study performed global phosphoproteomic analysis…”

Line 110: “proteomics studies”

Line 190:”Several in silico studies have looked at…”

Line 213:”tissu”

Line 218: “…independent functions is suggestive”

Some commas are missing, especially in sentences such as : Here, …. was presented…”

I believe the paper can be improved. The presented MS needs extensive work. However, in my opinion, minor revision is required

Kind regards

Author Response

Reviewer #3:

Author of this manuscript reviewed recent advances in the field of the role of MASTL regulation in cancer. The authors focused on the kinase-independent functions of MASTL and the potential therapeutic application of MASTL hampering. The introduction provides background and includes relevant references. This is very interesting and prevailing topic; however it is presented in a little  disorganized way.

Au: We thank the reviewer for their positive comments and we have addressed their concerns in the point-by-point response below.

Some of concerns are listed below:

It is not exactly clear, which parts of MS refer to regulation of MASTL in cancer cells and which describe mechanisms in normal cells, especially in lines 163-179. Cancer cell type/origin should addressed in some sentences (the statement “in cancer cells” is not enough).

Au: We have now mentioned specifically the cells type in the text (line 154, line 164, line 167, line 169, line 172, line 173, lines 177-178, line 179 and line 182).

In the section “AKT, Glycogen synthase kinase 3 (GSK3) and Wnt activity” – the meaning of the presented regulation is lost – what GSK3 inactivation accounted for? And conversely, what happened in tumour cell when GSK3 was activated? What was the cause of such different regulation? Cancer cell phenotype? Authors present some interesting findings, but no explanation is given. The findings should be better addressed – how the changes in signal transduction pathways accounted for cancer cell functioning (e.g. invasive properties), phenotype? The most essential parts are missing.

Au: We apologize for the obscurity of this chapter. The activation of GSK3 has been studied in three MASTL articles [11, 13 and 14]. GSK3 activation (decreased phosphorylation of GSK3a S21) has been shown to be responsible for the degradation of PHLPP phosphatase, which is responsible for ATK phosphorylation [14]. On the contrary, significant inactivation (increased phosphorylation of GSK3a S21) was observed in another study using the same cell type (MCF10A) [13], despite observing a similar invasive phenotype. In addition, the GSK3 activation after MASTL silencing was noted in colon cancer cells, resulting in B-catenin degradation [11]. In conclusion, the cancer proliferation was affected similarly by MASTL in all of these studies, but most likely GSK3 regulation is not the main mechanism involved, and may have context-dependent roles that are currently not fully understood. We have now modified this chapter to better clarify the disparity in the current literature.

Figure 2b left panel: since kinase-independent functions of MASTL in cancer was the main topic, and mechanistic studies were highlighted throughout the text, this illustration should better illustrate the presented findings. The main upstream regulators along with downstream effectors may be showed in detail. Alternatively, each mechanism the Authors describe, may be presented in a separate figure.

Au: We have updated figure 2 (old figure 1b) to mention the specific pathways and genes regulated by MASTL in a kinase-dependent and -independent manner. In addition, we have included a new figure depicting the possible scenarios of therapeutic targeting of MASTL, considering the kinase-dependent and -independent mechanisms (figure 4).

The manuscript needs editing and some attention to sentence structure so that the data are more clear to readers. Sections have a confusing numbering (there are two 2.1.1 sections, the first 2.1.1 section is vast and not divided into smaller fragments). Why did the authors create a separate paragraph 4.1. Figures and Tables? Figures and tables should be presented in the text, where necessary.

Au: We agree with the reviewer and have restructured the headings in line with the comments from both reviewers #2 and #3. This included the removal of the “4.1 Figures and Tables” section, as advised.

There are some shortcomings throughout the text:

Line 77: “Hyperactivation of AKT S473 has been proposed..”,

Au: We have edited this paragraph as suggested by reviewer #2, which included the removal of this sentence.

Line 84: “a slight increase of AKT S473 was observed”

Au: This sentence has now been revised, in line with the above comment and that of reviewer #2.

Line 215: „The organs (names?) written with (in?) pink…”

Au: We have adjusted the figure legend for Figure 1A (now figure 1B) to include the organ names that are highlighted in pink, as well as to correct “with” to “in”.

Please, give full name for all acronyms throughout the text, where listed for the first time, or, alternatively, put all abbreviations into one section (e.g. line  100 “cytoskeleton-associated proteins; namely Diap1, Dock1, Flna, Pak1-3, Pik3r1, Prkca, Raf1, Tln1, Zyx, Map2k2, Myh9, Wasf2, 101 Vasp, Vcl”; line 119: “NCMR ” was not defined)

Au: In line with the reviewers’ comments, we have now given full names for all acronyms in the text.

Please, use Italic for “via” “in vitro”, “in vivo” throughout the text (e.g. lines 202, 227, 232, 236, 238, Table 1).

Au: We have now corrected all instances of  “via”, “in vitro”, “in vivo” and “in silico” to italics throughout the text. 

Conclusions: some of the most important facts need to be better addressed. The main observations and findings cited  in previous paragraphs should be pointed out here. Since one of the main goals of the MS was to present how targeting of MASTL may be used for anti-cancer therapy, the most important findings in the area may be presented in conclusion section.

Au: To support the conclusions, we have included a new figure depicting the biological functions and possible benefits for therapeutic targeting of MASTL and the MASTL functions that are expected to be diminished by the different strategies.

The text need editing (two types of font were used).

Au: We have now corrected the discrepancy in the fonts, and the whole text is now in Palatino Linotype.

The English style and grammar should be improved, especially:

Line 36:”… the primary phosphorylation substrate of focal adhesion kinase seems to be itself”

Au: We have corrected “of” to “for”.

Line 49:”… The roles of MASTL in cancer are many and varied….”

Au: We have now corrected “many and varied” to “many and various”.

Line 82: “Another study performed global phosphoproteomic analysis…”

Au: We have now corrected “analysis” to “analyses”.

Line 110: “proteomics studies”

Au: We have now corrected “proteomics” to “proteomic”.

Line 190:”Several in silico studies have looked at…”

Au: We have now corrected “looked at” to “investigated the”.

Line 213:”tissu”

Au: We have now fixed the spelling for “tissue”.

Line 218: “…independent functions is suggestive”

Au: We have now corrected “is suggestive” to “are suggestive”.

Some commas are missing, especially in sentences such as : Here, …. was presented…”

Au: While we agree with the reviewer that “here” should often be followed by a comma, we believe that a reduced burden of commas throughout the text helps to give a smoother flow, and is helpful for the wider readership.

I believe the paper can be improved. The presented MS needs extensive work. However, in my opinion, minor revision is required

Au: We thank the reviewer for their detailed and helpful comments.

Round 2

Reviewer 2 Report

Accepted